# Interrogating endothelial barrier regulation by temporally resolved kinase network generation

Ling Wei[1], Selasi Dankwa[1], Kamalakannan Vijayan[1], Joseph D Smith[1,2,3], Alexis Kaushansky[1,2,3]

**Kinases are key players in endothelial barrier regulation, yet their temporal function and regulatory phosphosignaling networks are incompletely understood. We developed a novel methodology, Temporally REsolved KInase Network Generation (TREKING), which combines a 28-kinase inhibitor screen with machine learning and network reconstruction to build time-resolved, functional phosphosignaling networks. We demonstrated the utility of TREKING for identifying pathways mediating barrier integrity after activation by thrombin with or without TNF preconditioning in brain endothelial cells. TREKING predicted over 100 kinases involved in barrier regulation and discerned complex condition-specific pathways. For instance, the MAPK-activated protein kinase 2 (MAPKAPK2/MK2) had early barrier-weakening activity in both inflammatory conditions but late barrier-strengthening activity exclusively with thrombin alone. Using temporal Western blotting, we confirmed that MAPKAPK2/MK2 was differentially phosphorylated under the two inflammatory conditions. We further showed with lentivirus-mediated knockdown of MAPK14/p38α and drug targeting the MAPK14/p38α–MAPKAPK2/MK2 complex that a MAP3K20/ZAK-MAPK14/p38α axis controlled the late activation of MAPKAPK2/MK2 in the thrombin-alone condition. Beyond the MAPKAPK2/MK2 switch, TREKING predicts extensive interconnected networks that control endothelial barrier dynamics.**

## Introduction

Dysregulation of the blood–brain barrier is associated with the pathogenesis of a variety of diseases, including infectious causes such as cerebral malaria and noninfectious causes such as multifactorial neurodegenerative diseases (Miller et al, 2013; Daneman & Prat, 2015; Zhao et al, 2015). The permeability of endothelial barriers is tightly regulated via cell-cell junctions and focal adhesions. These points of cell–cell and cell–substrate contact are controlled by phosphorylation and actomyosin-mediated contractile

mechanisms, whose functions are heavily regulated by diverse phosphosignaling pathways (Mehta & Malik, 2006). Kinases and phosphatases can rapidly (on the scale of seconds to minutes) alter phosphorylation states in response to barrier-disruptive stimuli and can promote long-term regulatory functions during barrier recovery (Kuppers et al, 2014). Although kinase signaling pathways mediating barrier function have been intensively explored, there is an incomplete understanding of the temporal regulation of these pathways and their complex network-level connectivity (Mehta & Malik, 2006; Kuppers et al, 2014; Komarova et al, 2017; Dankwa et al, 2021). Moreover, it is difficult to assign specific barrier activity to individual kinases because some kinases are activated in response to both barrier-strengthening and barrier-weakening mediators and/or display distinct time-dependent barrier functions after activation (Garcia et al, 2001; Vouret-Craviari et al, 2002; McVerry & Garcia, 2004; Knezevic et al, 2009; Birukova et al, 2013; Han et al, 2013; Klomp et al, 2019). Thus, understanding the temporal dynamics of barrier regulation and crosstalk between pathways is critical for designing successful targeted therapeutic interventions.

During altered states like infection or disease, endothelial cells must integrate complex inflammatory signals. For instance, thrombin and the proinflammatory cytokine TNF can both promote morphological remodeling of the endothelium that leads to the loss of barrier integrity (Marcos-Ramiro et al, 2014; Oldenburg & de Rooij, 2014). TNF exacerbates thrombin-induced barrier disruption (Anrather et al, 1997; Tiruppathi et al, 2001; Liu et al, 2004), which indicates that endothelial signaling pathways and/or their kinetics are altered by combined inflammatory stimuli. High-resolution and time-resolved phosphoproteomic analyses have cataloged phosphorylation events that occur in response to thrombin or TNF-induced endothelial activation (van den Biggelaar et al, 2014; Beguin et al, 2019). Separate efforts have developed computational tools and modeling approaches that use omics data for network generation, including logic modeling such as Boolean logic models and logic ordinary differential equations, enrichment-based approaches such as kinase–substrate enrichment analysis (Casado et al, 2013; Vaga et al, 2014; Schafer et al, 2019), and a network

[1]Center for Global Infectious Disease Research, Seattle Children's Research Institute, Seattle, WA, USA [2]Department of Pediatrics, University of Washington, Seattle, WA, USA [3]Department of Global Health, University of Washington, Seattle, WA, USA

Correspondence: alexis.kaushansky@seattlechildrens.org
Kamalakannan Vijayan's present address is School of Biology, Indian Institute of Science Education and Research, Thiruvananthapuram, India

propagation method that integrates kinase inhibitor screens with known protein–protein interactions (Bello et al, 2021). However, these computational tools rely on large-scale phosphoproteomics data that are infrequently collected with fine temporal resolution. Proteomics also only indirectly assesses kinase functionality, as inferred by phosphorylation states, and many kinases play multiple roles upon activation. Given these limitations, existing network generation tools have not yet been applied to elucidating the time-resolved molecular functional underpinnings of endothelial barrier integrity.

Kinase regression (KiR) is an approach based on a compound screen that uses a small panel of kinase inhibitors with overlapping target specificities and implements elastic net regression to broadly interrogate the function of hundreds of kinases in a cellular phenotype (Gujral et al, 2014; Arang et al, 2017; Dankwa et al, 2021). By integrating KiR predictions with known kinase–substrate phosphorylation databases, kinase-centered functional networks important for particular phenotypes can be built (Bello et al, 2021). Kinase inhibitors can alter the temporal features of endothelial barrier disruption and barrier recovery after thrombin stimulation (Dankwa et al, 2021), suggesting that different sets of kinases are functional at different stages of barrier perturbation. Here, we develop a novel methodology, Temporally REsolved KInase Network Generation (TREKING), and demonstrate its utility to investigate time-resolved kinase functionality associated with barrier regulation of human brain microvascular endothelial cells (HBMECs) and to build phosphosignaling network models of endothelial barrier regulation.

# Results

### TNF preconditioning alters kinase determinants of thrombin-induced barrier perturbations

To investigate kinase signaling pathways induced by thrombin and explore how the phosphosignaling environment may be altered by sequential inflammatory stimuli, we exploited our published KiR dataset of HBMEC monolayers stimulated with thrombin, with or without TNF preconditioning (Dankwa et al, 2021). Thrombin causes an acute drop in HBMEC barrier integrity that peaks around 20–30 min and a slower barrier recovery over the next 1.5 h (Figs 1A and S1A and B). TNF preconditioning exacerbated thrombin-induced barrier disruption, as demonstrated by the increased maximum barrier permeability and prolonged barrier recovery (Figs 1A and S1A and B). Although this finding is consistent with previous reports (Anrather et al, 1997; Tiruppathi et al, 2001; Liu et al, 2004), the molecular networks that regulate these differences remained unknown.

To define kinase signaling mechanisms underlying TNF potentiation of thrombin-induced barrier disruption, we reanalyzed the dataset from a small chemical compound screen against human kinases. In the original KiR analysis, HBMEC monolayers were treated with thrombin (–/+TNF preconditioning), and 28 kinase inhibitors were added 6 min after thrombin treatment (Fig 1B) (Dankwa et al, 2021). The endothelial barrier integrity was assessed in real-time using the xCELLigence system, which measures cellular

impedance across adherent cells in electronic microplate wells. Decreases in the cell index correlate with the initial strong retraction of endothelial cells after thrombin treatment during the barrier disruption phase, whereas increased cell index values are associated with recovery of monolayer integrity (Dankwa et al, 2021). We observed that the DMSO vehicle alone had no effect on HBMECs treated with thrombin or TNF (Fig S1C) but that different kinase inhibitors blunt, have minimal effect, or exacerbate thrombin-induced barrier disruption and exhibit temporal features during the 6-h time window (Fig 1B) (Dankwa et al, 2021).

We reasoned that if similar kinases control barrier integrity in the two inflammatory conditions within a given time window, there should be a stronger correlation between the activities of the 28 kinase inhibitors in thrombin-alone and TNF-preconditioned settings in that temporal period. In contrast, if different kinases control the barrier integrity in the two conditions, we expect less correlation between the activities of the kinase inhibitors. Thus, the KiR screen can be used to probe the underlying barrier regulatory phosphosignaling networks across conditions. Using area under the curve (AUC) of the normalized cell index as the metric, we compared the two inflammatory conditions across four different 10-min time windows: barrier disruption (6–16 min after thrombin treatment), early barrier recovery (31–41 min after thrombin treatment), mid barrier recovery (121–131 min after thrombin treatment), and late barrier recovery (241–251 min after thrombin treatment) (Fig 1C). We also evaluated the correlation between the activities of the 28 kinase inhibitors in +thrombin and TNF preconditioning+ thrombin conditions within a 5-min time window that slides at 1-min steps for the first 2 h after kinase inhibitor treatment and at 5-min steps afterward (Fig 1D). The correlation was the lowest in the early barrier disruption phase (Pearson's r = 0.72) and increased during early barrier recovery (Pearson's r = 0.90 to 0.92) (Fig 1D), suggesting that the phosphosignaling networks are more divergent during barrier disruption and more similar during the initial barrier restoration phase. Of note, differences in the kinetics of action of the kinase inhibitors may also contribute to the reduction of the correlation at the earliest time points. Notably, the correlation declined again at ~120 min and remained divergent throughout the entire mid-to-late barrier recovery phases (Pearson's r = 0.91 to 0.77), a period when the two inflammatory conditions diverged in their slopes of recovery and final cell index (compare Fig 1A and D). This analysis is consistent with a model where TNF preconditioning alters kinase regulatory networks that dictate the extent and kinetics of thrombin-induced barrier disruption and recovery in brain endothelial cells.

### Time-resolved predictions of barrier-weakening and barrier-strengthening kinases between the two inflammatory conditions

Previously, we predicted 29 and 25 kinases that regulate the HBMEC barrier in response to thrombin treatment with or without TNF preconditioning, respectively (Dankwa et al, 2021). However, these KiR predictions were based on the full AUC (6-h time window, post-thrombin treatment) and did not consider time windows of barrier activity. To address this lack of temporal resolution, we performed

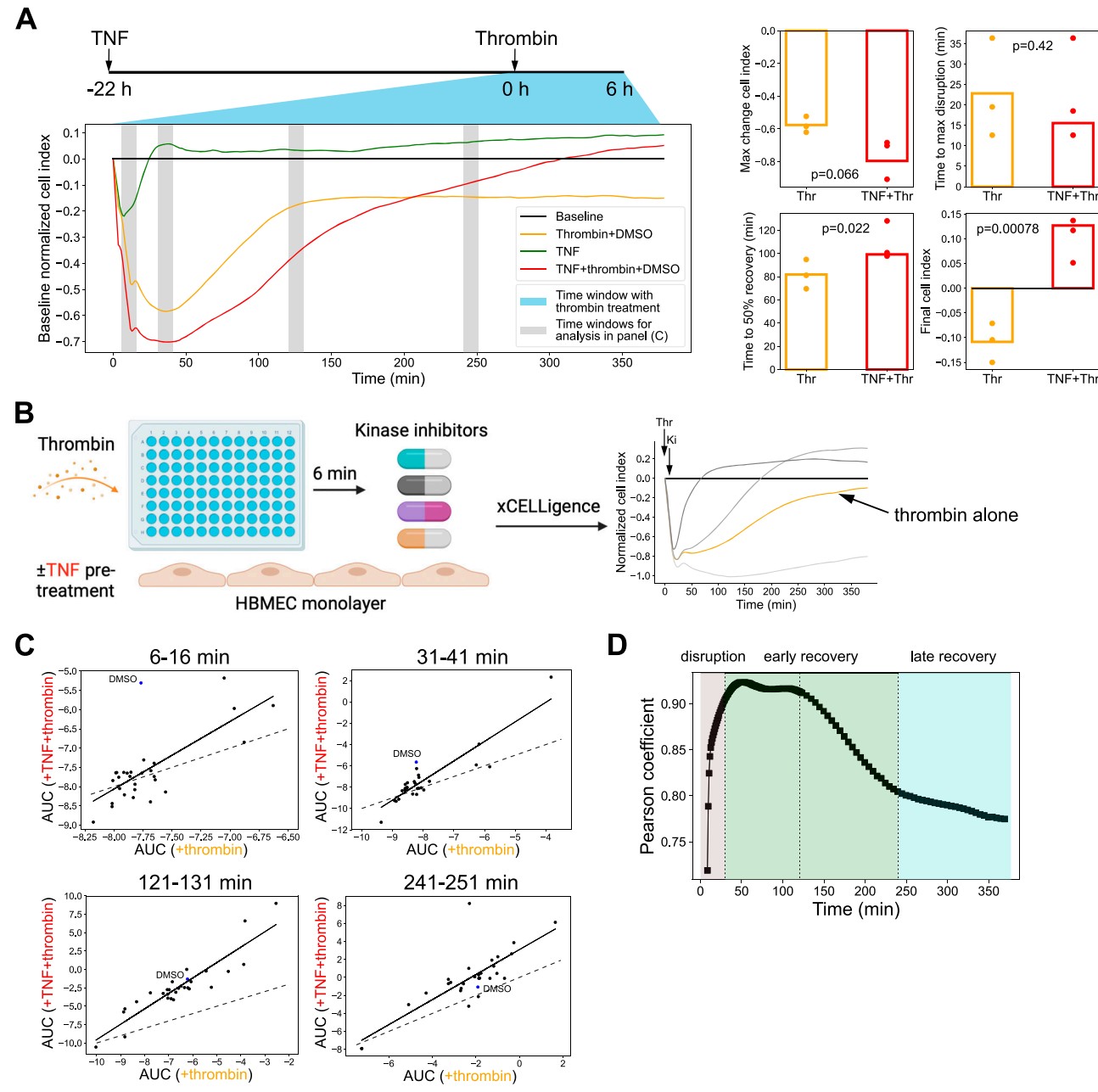

**Figure 1. Preconditioning with TNF alters the temporal kinetics and magnitude of the thrombin-induced barrier disruption response.**
**(A)** Left: representative xCELLigence data showing barrier response to thrombin with or without TNF preconditioning. Right: Quantification of the maximum change and time to 50% recovery (N = 3). *P*-values from the *t* test are shown. **(B)** Graphic of the KiR screen conducted with 28 kinase inhibitors for their impact on barrier permeability in thrombin-treated HBMECs (−/+TNF preconditioning). Monolayers of HBMECs were treated with thrombin followed at 6 min by kinase inhibitor treatment, and barrier permeability was measured in real-time using the xCELLigence system. The xCELLigence curves on right illustrate kinase inhibitor changes in the thrombin-induced barrier response (reprinted from Fig 2F of Dankwa et al [2021]). **(C)** Correlations of the 28 kinase inhibitors between +thrombin and TNF preconditioning+thrombin conditions at different stages of barrier perturbation. The area under the corresponding permeability curve (AUC) within representative time windows (10 min) is displayed as scatter plots (blue circle: DMSO-treated; black circles: kinase inhibitor-treated; black solid line: linear fitting line excluding the DMSO control; black dashed line: identity line). The corresponding time windows to the scatter plots are indicated in panel (A) with gray shading. **(D)** Pearson correlation coefficients between AUC in +thrombin and TNF preconditioning+thrombin conditions. Temporal AUC was calculated within a 5-min sliding time window that slides at 1-min steps for the first 2 h of kinase inhibitor treatment and at 5-min steps afterward. See also Fig S1.

temporal KiR (tKiR) using a sliding-window analysis (Fig S2, top, schematic overview). For this approach, a 5-min sliding time window was applied to the normalized cell index data, which slides at 1-min steps for the first 2 h after kinase inhibitor treatment and at

5-min steps afterward (Figs 2A and S2). Among the 291 protein kinases used for tKiR predictions (Anastassiadis et al, 2011) (Fig S2, see methodology), the number of functional kinases increased to 120 and 108 in HBMECs treated with thrombin −/+TNF preconditioning,

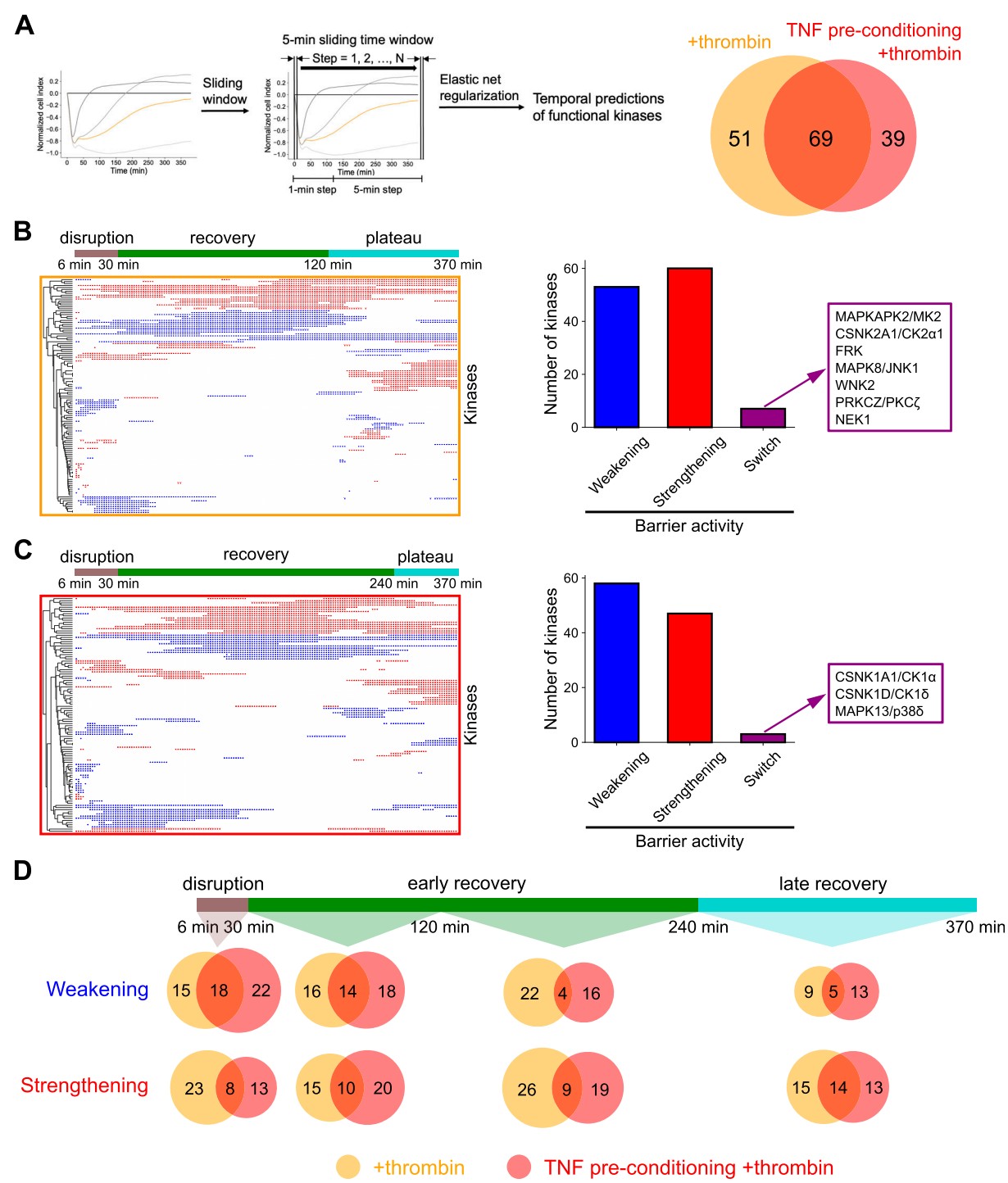

**Figure 2. Using kinase regression to inform kinases important for barrier regulation with temporal resolution.**
**(A)** Left: schematic describing the tKiR methodology to predict time windows of kinase barrier activity. A 5-min sliding time window was applied to the normalized cell index data from the xCELLigence assays conducted with the 28 kinase inhibitors. Right: Venn diagram showing the total number of kinases predicted in each condition across all time windows. **(B, C)** Left: Time-resolved barrier activity profiles of predicted kinases in (B) +thrombin and (C) TNF preconditioning+thrombin conditions. Blue and red colors represent barrier-weakening and barrier-strengthening functionality, respectively. Predicted kinases are hierarchically clustered using the Euclidean distance metric. Right: Number of kinases predicted to be barrier-weakening, barrier-strengthening, or of dual functionality across the time course ("switch kinases"). **(D)** Venn diagrams showing the number of kinases predicted to be barrier-weakening or barrier-strengthening at different stages of barrier perturbation. See also Table S1.

respectively (Fig 2A; Table S1). The higher resolution of tKiR is because the sliding-window analysis allows for the detection of kinases whose activity is brief or sporadic. In total, 159 kinases (~30% of the total human kinome) were predicted to regulate barrier function within the 6-h timeframe, with 69 kinases being common to the two conditions (Fig 2A).

As a further refinement, tKiR models were used to predict the kinase functionality (promoting barrier weakening or promoting barrier strengthening) in a time-resolved manner for the two inflammatory conditions (Fig 2B and C). The barrier activity of individual kinases was inferred by the linear regression slope of the kinase inhibitors targeting that kinase from the KiR analysis, where barrier-weakening or barrier-strengthening kinases are defined as the kinases that promote disruption or strengthening of barrier integrity, respectively (Fig S2, top). For HBMECs not preconditioned with TNF, 53 and 60 kinases were predicted to play a barrier-weakening and barrier-strengthening role, respectively (Fig 2B). In addition, seven kinases were predicted to play both barrier-weakening and barrier-strengthening roles during different time windows and were termed "switch kinases" (Fig 2B). For HBMECs preconditioned with TNF, 58 and 47 kinases were predicted, respectively, to be barrier-weakening and barrier-strengthening, and three switch kinases were predicted (Fig 2C). Overall, a greater number of barrier-weakening kinases were predicted during the initial disruption phase, whereas the proportion of barrier-strengthening kinases increased over barrier recovery (Fig 2D). Nevertheless, cells propagate both types of barrier activities concurrently, and it is the balance that changes over time (Fig 2D).

### The tKiR predictions of kinase activity recapitulate information flow through canonical MAPK signaling pathways involved in barrier regulation

Previous work has established that MAPK cascades are activated after thrombin stimulation, including the extracellular signal-regulated kinases (ERKs), Jun N-terminal kinases (JNKs), and p38 MAPKs (Minami et al, 2004; Mehta & Malik, 2006; Radeva & Waschke, 2018). The topology of MAPKs is well-established; cell surface receptors such as receptor tyrosine kinases (RTKs) provide a link between extracellular stimuli and the cascade (Morrison, 2012). After activation, most MAPK pathways have a four-tier kinase architecture (MAP3K-MAP2K-MAPK-MAPKAPK) (Pimienta & Pascual, 2007). Given the strict directionality of the network, we reasoned that RTKs would be predicted to regulate the barrier during the earliest time points after thrombin treatment, followed sequentially by MAP3Ks, MAP2Ks, MAPKs, and finally MAPKAPKs. Furthermore, kinases involved in a signaling cascade would be more likely to have the same barrier activity over a given time window, especially if they were acting to propagate a specific barrier phenotype.

Overall, the ERK, JNK, and p38 pathways were all implicated in thrombin signaling, with multiple kinases in each signaling cascade predicted by tKiR (Fig 3). Specifically, within the ERK and the JNK pathways, most of the predicted kinases were barrier-weakening, and the windows when their activity was predicted followed the expected order of MAP3Ks or MAP2Ks preceding MAPKs, which preceded downstream MAPKAPKs (Fig 3). Although the ERK pathway had both an early and a late wave of barrier-weakening activity, the JNK pathway was predicted to have primarily early barrier activity (Fig 3). By comparison, the p38 pathway had a mixture of early barrier-weakening activity followed by a switch to late barrier-strengthening activity, which was mediated by different p38

isoforms (Fig 3). For instance, MAPKAPK2/MK2 was one of the seven switch kinases in the +thrombin condition (Fig 2B) and had an early barrier-weakening activity and late barrier-strengthening activity (Fig 3). MAPKAPK2/MK2 is a primary target of MAPK14/p38α, although it can also be activated by MAPK1/ERK2 and JNK signaling (Ronkina et al, 2008; Johnson et al, 2023). Of interest, a mid-to-late–acting barrier-strengthening p38 pathway could be assembled from known components, consisting of MAP3K20/ZAK-MAPK14/p38α-MAPKAPK2/MK2, but different upstream kinases appeared to be involved in the early barrier activity of MAPKAPK2/MK2 under thrombin stimulation (Fig 3). Thus, our tKiR model predicted different MAPK kinases were responsible for early and late periods of MAPKAPK2/MK2 activation.

The same overall MAPK pathways were predicted with TNF preconditioning; however, the timing and duration of the ERK and the JNK pathways were shifted earlier and shortened for most kinases (Fig 3). TNF preconditioning also modified barrier-regulatory signaling cascades within the three canonical MAPK pathways. In the ERK pathway, there was a second wave of barrier-weakening activity by MAPK1/ERK2, and in the JNK pathway, the barrier-weakening activity of TAOK1 was also shifted from an early to a later timeframe. Of interest, TAOK1 can promote activation of the ERK1/2 pathway in macrophages (Zhu et al, 2020), suggesting that TAOK1 may also function in the ERK pathway in regulating barrier properties. In the JNK pathway, the number of barrier-regulatory kinases was increased, consistent with previous work indicating that combined TNF and thrombin treatment alters the magnitude and duration of JNK activation in endothelial cells (Liu et al, 2004). In the p38 pathway, the barrier activity of the p38 isoforms was rewired. Specifically, although MAPK13/p38δ retained a brief barrier-weakening activity, the late barrier-strengthening MAPK14/p38α-MAPKAPK2/MK2 pathway was absent and other p38 isoforms replaced it. Therefore, we termed MAPKAPK2/MK2 a condition-specific, switch kinase. It was predicted to have early barrier-weakening activity in both inflammatory conditions but late barrier-strengthening activity only in the thrombin-alone condition. Consequently, this analysis suggests that core MAPK signaling pathways involved in thrombin-induced barrier disruption remain intact after TNF preconditioning, but the timing, duration, and importance of specific MAPKs are altered.

### Kinase activation state is largely consistent with tKiR-predicted barrier activity in the ERK and the JNK pathways

To investigate the tKiR predictions about MAPK signaling pathways regulating barrier phenotypes in the two inflammatory conditions, we probed kinase phosphorylation over time via Western blot. We reasoned that there should be a rise in phosphorylation at kinase activation sites immediately preceding and/or during periods of predicted barrier activity. For this analysis, HBMECs (−/+TNF preconditioning) were treated with thrombin for 0, 5, 15, 30, 60, 120, 180, 240, or 360 min. In each experimental condition, the level of phosphorylated kinase after thrombin treatment was compared with its phosphorylation at the basal level (media only) after normalization to GAPDH. Initially, we probed the ERK and JNK pathways that were primarily predicted to be involved in barrier-disruptive signaling cascades from the tKiR analysis. Control Western blots showed that the total levels of the ERK1/2

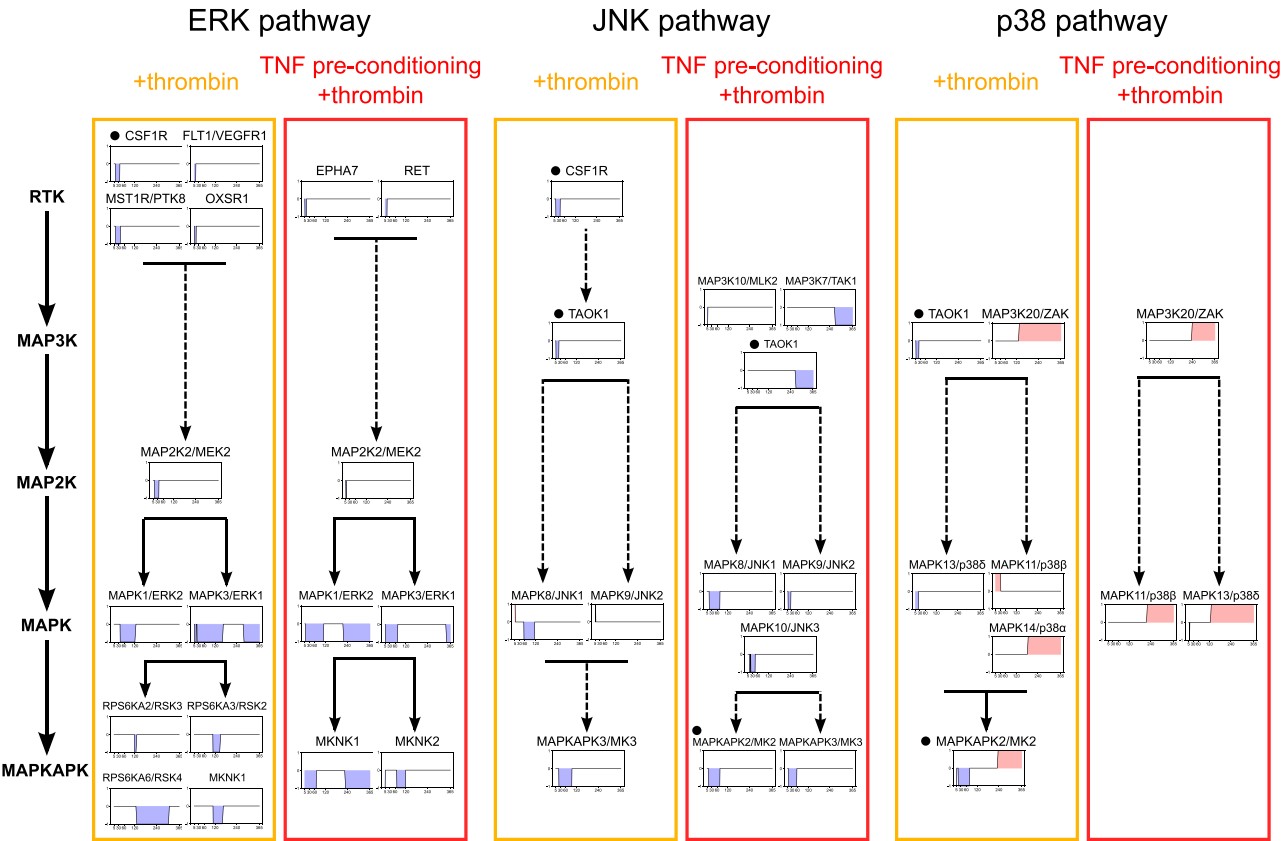

**Figure 3. tKiR methodology recapitulates the phosphosignaling events in canonical MAPK pathways.**
Members of three canonical MAPK pathways (ERK, JNK, and p38) were predicted to regulate barrier function in thrombin-treated HBMECs with or without TNF preconditioning. Kinases were mapped onto the corresponding signaling modules along with their temporal functionality profiles, where blue and red shadings represent barrier-weakening and barrier-strengthening activity, respectively. Solid arrows represent direct kinase phosphorylation interactions reported previously, and dashed arrows represent indirect interactions as reported in the previous literature (Cargnello & Roux, 2011; Ronkina & Gaestel, 2022). Kinases overlapping between different MAPK pathways are marked by a black dot by the kinase name. See also Table S1.

and JNK1/2 isoforms did not change in this period (Fig S3A) and therefore did not change the normalization of phosphorylated protein levels compared with normalization to GAPDH (Fig S3B). Within the canonical ERK cascade, there was a sequential activation of kinases and statistically significant increase in activation-associated phosphorylation in MAP2K1/2 (MEK1/2), MAPK1/ERK2, MAPK3/ERK1, and the downstream MAPKAPK target MKNK1 prior or during predicted early barrier-weakening activity in both inflammatory conditions (Fig 4A). Likewise, there was a second wave of MAPK1/ERK2 and MAPK3/ERK1 late barrier activity from 180 min to 360 min after thrombin treatment in both inflammatory conditions that was independent of late MAP2K1/2 (MEK1/2) activation (Fig 4A), as predicted by the tKiR model (Fig 3). As MAP2K1/2 (MEK1/2) were only predicted to have early barrier activity (Fig 3), this raises the possibility that another kinase(s), such as TAOK1, is responsible for the late MAPK1/3 (ERK2/1) activation or there is a change in the activity of a phosphatase(s) that can dephosphorylate MAPK1/3 (ERK2/1).

Results from temporal Western blot analysis also showed that activation marks increased in both MAPK8/JNK1 and MAPK9/JNK2 before the predicted early barrier-weakening activity in both inflammatory conditions (Fig 4B). Moreover, the timing, magnitude, and duration of early MAPK8/JNK1 activation increased with TNF

preconditioning, consistent with a shift in the timing of barrier activity predicted by tKiR (Fig 4B). There were also some notable discrepancies between activation marks and predicted barrier activity. In +thrombin condition, we only observed a very modest and statistically significant increase in the activity of MAPK8/JNK1 and MAPK9/JNK2 at early time points where they were predicted to have barrier-strengthening activity (Fig 4B). In addition, there was an increase in MAPK8/JNK1 and MAPK9/JNK2 activation marks at late time points (180 and 240 min after thrombin treatment) in +thrombin condition (Fig 4B), despite the lack of predicted barrier activity. One possibility is that the late JNK1/JNK2 activation marks may be related to a non-barrier function of JNK kinases. Alternatively, this may be a false-negative tKiR prediction. Overall, in most cases, increasing phosphorylation levels were observed immediately preceding or overlapping with the corresponding time windows of tKiR-predicted barrier activity for the ERK and the JNK pathways.

Our tKiR model predicted that MAPKAPK2/MK2 had a complex barrier activity that differed between the two inflammatory conditions. In the thrombin-alone condition, MAPKAPK2/MK2 was predicted to be a switch kinase with early barrier-weakening and late barrier-strengthening activity. However, the late barrier-strengthening activity was predicted to be disconnected after

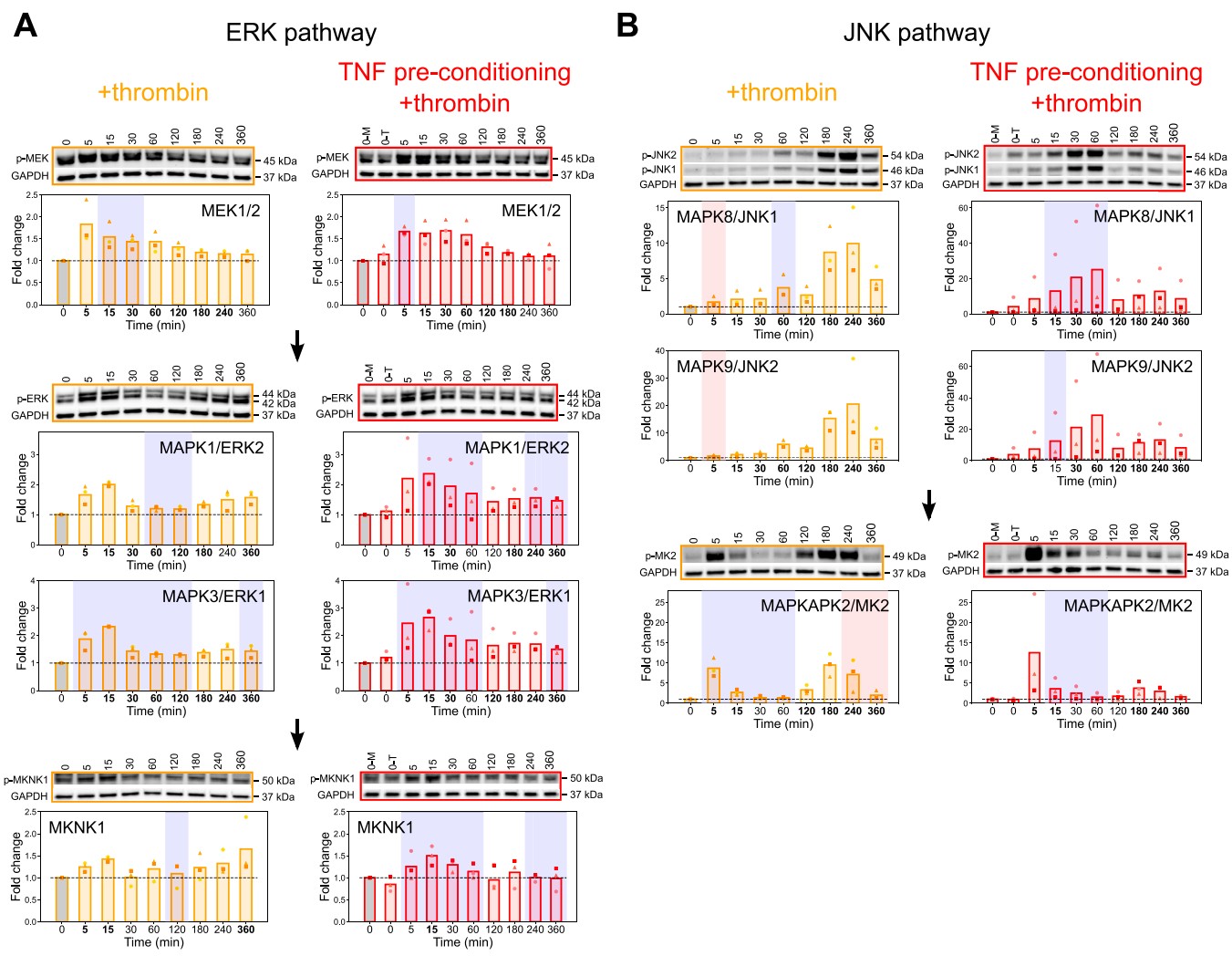

**Figure 4. Correlation between phosphorylation activation marks in ERK and JNK kinases and tKiR-predicted barrier activity.**
**(A, B)** Phosphorylation of members in (A) ERK and (B) JNK pathways was probed by Western blot in three biological replicates. To study the correlation between kinase activation marks and predicted barrier activity, HBMECs were stimulated with TNF (10 ng/ml) for 21 h or left unstimulated, and cells were then treated with thrombin (5 nM) for the indicated times. Cells preconditioned with TNF but not treated with thrombin are labeled "0-T." Data were normalized to non-treated, media-only condition ("0-M," gray bars). GAPDH was used as a loading control. Symbols represent the fold change of individual biological replicates, and bars represent the mean fold change of three biological replicates. Blue and red shadings represent, respectively, barrier-weakening and barrier-strengthening activity, as predicted by tKiR. The bolded timepoints indicate that the phosphorylation is different from the basal level ($P$-value from the $t$ test below 0.05 or all biological replicates reporting a fold change increasing by at least 20% compared with non-treated cells). See also Fig S3 and Table S3.
Source data are available for this figure.

TNF preconditioning (Fig 3). As MAPKAPK2/MK2 can be activated by ERK, JNK, and p38 pathways, we used Western blots to investigate how the three MAPKs signaling pathways may regulate this complex barrier phenotype. Consistent with the tKiR model (Fig 3), MAPKAPK2/MK2 activation increased at both early and late time points in +thrombin condition, but the late MAPKAPK2/MK2 activation was substantially reduced with TNF preconditioning (Fig 4B). The early MAPKAPK2/MK2 activation correlated with a rise in MAPK1/ERK2 and MAPK3/ERK1 activation (Fig 4A), and the late MAPKAPK2/MK2 activation correlated with a rise in MAPK8/JNK1 and MAPK9/JNK2 activation (Fig 4B). Moreover, the late MAPK8/9 (JNK1/2) activation marks were substantially reduced after TNF preconditioning (Fig 4B). This analysis indicates that the ERK pathway

is more likely to contribute to early MAPKAPK2/MK2 activation, whereas the JNK pathway is more likely to contribute to late MAPKAPK2/MK2 activation.

### Differential p38-driven activation of the MAPKAPK2/MK2 switch kinase in the two inflammatory conditions

To further investigate the MAPKAPK2/MK2 switch kinase, we examined the p38 pathway. The tKiR methodology predicted that a MAP3K20/ZAK-MAPK14/p38α-MAPKAPK2/MK2 pathway was involved in late barrier strengthening in +thrombin condition and that this pathway is disconnected with TNF preconditioning (Figs 3 and 5A). To test the model predictions, we probed the

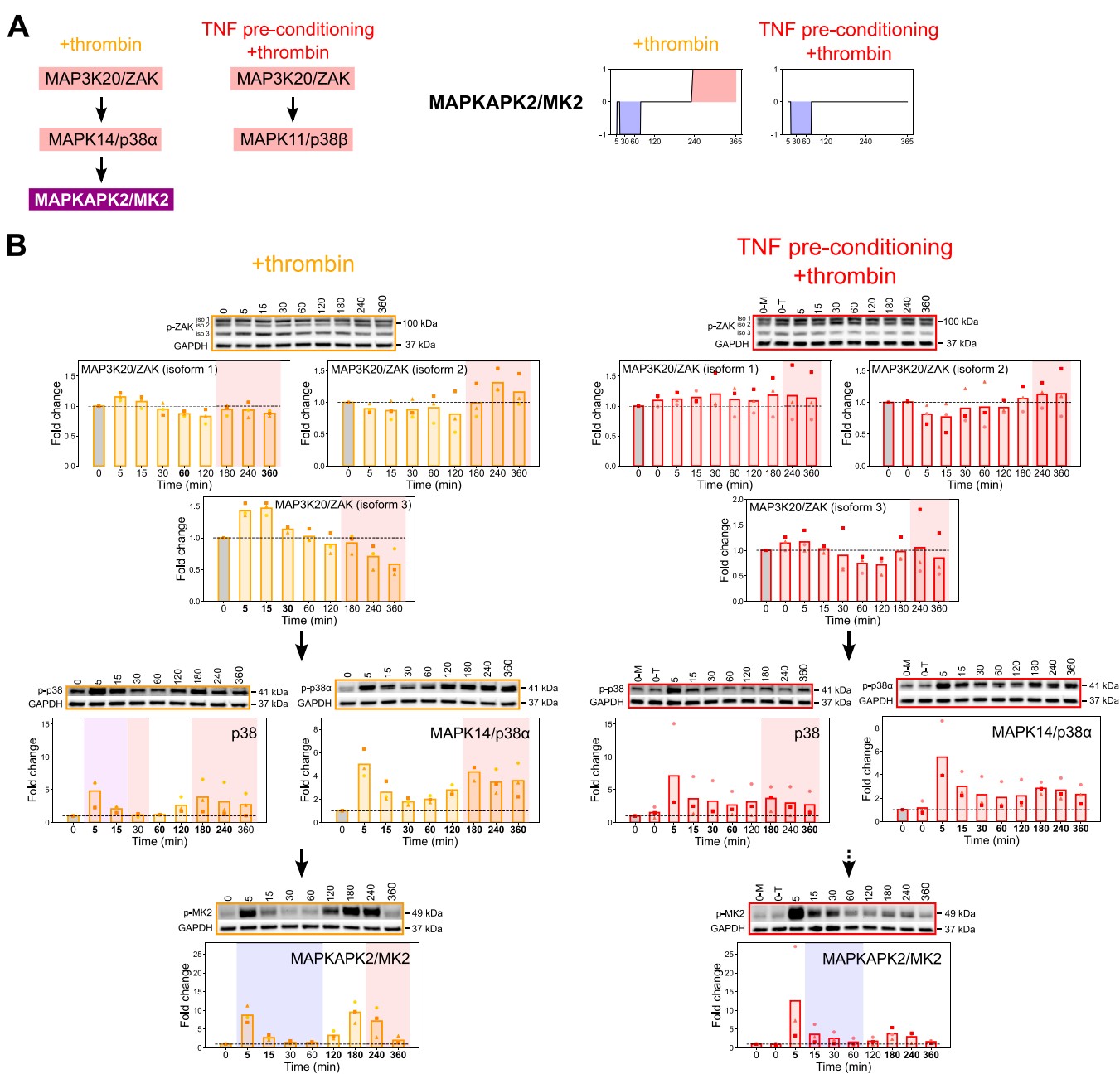

**Figure 5. Differential p38-driven activation of the MAPKAPK2/MK2 switch kinase.**
**(A)** Left: tKiR model of how TNF preconditioning rewires the late barrier activity of the p38 pathway. Right: Temporal barrier activity of MAPKAPK2/MK2 as predicted by tKiR. **(B)** HBMECs were stimulated with TNF (10 ng/ml) for 21 h or left unstimulated, and cells were then treated with thrombin (5 nM) for the indicated times. Phosphorylation of members in the p38 pathway was probed by Western blot in three biological replicates. Cells preconditioned with TNF but not treated with thrombin are labeled "0-T." Data were normalized to non-treated, media-only condition ("0-M," gray bars). GAPDH was used as a loading control. Symbols represent the fold change of individual biological replicates, and bars represent the mean fold change of three biological replicates. Blue and red shadings represent, respectively, barrier-weakening and barrier-strengthening activity, as predicted by tKiR, and purple shadings represent instances where different p38 kinases with barrier-weakening or barrier-strengthening functionality were predicted at that time point. The bolded timepoints indicate that the phosphorylation is different from the basal level (*P*-value from the *t* test below 0.05 or all biological replicates reporting a fold change increasing by at least 20% compared with non-treated cells). See also Fig S3 and Table S3. Source data are available for this figure.

phosphorylation levels of MAP3K20/ZAK and the p38 isoforms by Western blot. Control Western blots showed that the total levels of p38 isoforms did not change in this period (Fig S3A). The *MAP3K20/ZAK* gene is alternatively spliced into large (~100 kD) and small (~55 kD) isoforms (Gotoh et al, 2001). In HBMECs, phosphorylated

MAP3K20/ZAK ran as two large isoforms (labeled isoforms 1 and 2) and a smaller isoform (labeled isoform 3) (Fig 5B). Although the smaller MAP3K20/ZAK isoform 3 (~70 kD) was activated at early time points (5 and 10 min after thrombin treatment), the larger MAP3K20/ZAK isoform 2 was activated at late time points (240 and 360 min

after thrombin treatment), coincident with a period of predicted MAP3K20/ZAK barrier-strengthening activity in both inflammatory conditions (Fig 5B). Likewise, MAPK14/p38α was activated at both early and late time points in +thrombin condition, but the late activation was dampened with TNF preconditioning (Fig 5B). Taken together, these findings suggest that the smaller MAP3K20/ZAK isoform 3 is likely responsible for the early p38 activation and the larger MAP3K20/ZAK isoform 2 for the late barrier-strengthening functionality of MAPK14/p38α-MAPKAPK2/MK2. Furthermore, the dampening of MAPK14/p38α and MAPKAPK2/MK2 activation in the late barrier recovery phase upon TNF preconditioning is consistent with the tKiR model predictions that sequential treatment with TNF and thrombin led to rewiring of this late-stage barrier-strengthening function to other p38 isoforms that are unable to activate MAPKAPK2/MK2. Consequently, there is a dampening of both late JNK (Fig 4B) and p38α activation (Fig 5B) after TNF preconditioning, which could contribute to the reprogramming of the MAPKAPK2/MK2 switch kinase.

## Construction of phosphosignaling networks that regulate barrier function

Among the kinases predicted by tKiR, there are many non-MAPK kinases (Table S1). However, much less is known about non-MAPK signaling cascades, and many kinases have been rarely studied. To discover new signaling pathways and crosstalk between pathways, we reasoned that kinases that are temporally related in barrier activity are more likely to act within a connected signaling module. Thus, we used self-organizing maps (SOMs) to systematically group kinases according to their temporal barrier activity. SOM is a dimensionality reduction method that can capture the topographic relationships of time series data (Kohonen, 1982). Kinases of similar temporal functionality were clustered into one group (called a "neuron"), and neurons having similar mean temporal behavior are more closely located on the SOM (Figs 6A, S2 [bottom] and S4A and Tables S2 and S3). To generate the SOMs, we used a grid size of 6 × 6 (resulting in up to 36 neurons). In +thrombin condition, kinases were grouped into 30 neurons, and each neuron contained between 1 and 11 kinases (Figs 6A and S4B; Table S2). In TNF preconditioning+thrombin condition, kinases were grouped into 35 neurons, and each neuron contained between 1 and 9 kinases (Figs 6A and S4B; Table S2).

To investigate systems-level interconnections, we built TREKING models of barrier phosphosignaling networks (Fig S2, bottom). Specifically, for each neuron, a local phosphosignaling network was built to describe the paths through which phosphosignals may propagate, using the kinase–substrate phosphorylation database PhosphoSitePlus (Hornbeck et al, 2015) to predict upstream and downstream kinases and to infer intermediate kinases in the pathways (Fig S4C). We were able to build 26 local networks from the 30 neurons generated in +thrombin condition and 27 local networks from the 35 neurons generated in TNF preconditioning+thrombin condition (Table S2). The remaining neurons included only a single kinase that does not self-phosphorylate or multiple unconnected kinases. The size and topology of the phosphosignaling networks varied across neurons. In +thrombin condition, most networks had a maximum node-to-node shortest path length below 10 kinases,

whereas the deepest networks built from neurons (0,0) and (3,3) had a maximum shortest path length of 14 kinases (Fig S4C). In TNF preconditioning+thrombin condition, most networks had a maximum shortest path length between 3 and 9 kinases, except for the network built from neuron (2,4) that had a maximum shortest path length of 16 kinases (Fig S4C).

## TREKING maps alternate routes for activation of the MAPKAPK2/MK2 switch kinase

We used the TREKING models to further investigate the regulation of the MAPKAPK2/MK2 switch kinase. In the +thrombin condition, MAPKAPK2/MK2 was clustered within a mid-to-late barrier-strengthening neuron (5,5). This neuron contained the MAP3K20/ZAK, MAPK14/p38α, and MAPKAPK2/MK2 components in the p38 pathway and G-protein–coupled receptors 3 and 4 (GRK3 and GRK4), C-terminal Src kinase (CSK; a negative regulator of Src-family kinases), and two members of noncanonical NF-κB signaling pathways (MAP3K14/NIK and inhibitor of nuclear factor kappa B kinase subunit beta [IKBKB/IKKβ]) (Fig 6B). In addition to the canonical MAP3K20/ZAK-MAPK14/p38α-MAPKAPK2/MK2 pathway (Figs 3 and 5), the TREKING model predicted several alternative pathways for late MAPKAPK2/MK2 activation (Fig 6C). One alternative pathway was predicted to link MAP3K20/ZAK and MAPK14/p38α via checkpoint kinase 2 (CHEK2), TTK protein kinase (TTK), ABL proto-oncogene 1 non-RTK (ABL1/c-Abl), and zeta chain of T-cell receptor–associated protein kinase 70 (ZAP70), instead via the canonical MAPK pathway topology (MAP3K-MAP2K-MAPK) (Fig 6C). Other routes included a CSK–LCK pathway and a noncanonical NF-κB signaling pathway involving MAP3K14/NIK and the inhibitor of nuclear factor kappa B kinase subunit alpha (CHUK/IKKα) and MAPK1/ERK2 (Fig 6C). Consistent with TREKING model predictions, activation marks increased in MAPK1/ERK2, ABL1/c-Abl, and CHUK/IKKα by Western blot in the late barrier recovery phase (Fig 6D). The TREKING model also predicted that activated MAPKAPK2/MK2 leads to activation of MAP3K5/ASK1, suggesting a potential positive feedback loop, via AKT serine/threonine kinase 1 (AKT1), which exhibited increased phosphorylation in the late barrier recovery phase (Fig 6C and D).

MAPKAPK2/MK2 was no longer predicted to be a switch kinase with TNF preconditioning and clustered with an early barrier-weakening neuron (5,3) that also included MAPK3/ERK1, MKNK1, MAPK8/JNK1, p21 (Rac1) activated kinase 3 (PAK3), and Janus kinase 1 (JAK1) (Fig S5A). To better understand the early barrier-weakening activity of MAPKAPK2/MK2, we built a composite phosphosignaling network from neurons (5,3) and (5,4) (Fig S5B). This TREKING model showed that ERK- and JNK-associated signaling may contribute to the activation of MAPKAPK2/MK2 in the barrier disruption phase and revealed crosstalk between ERK and JNK signaling pathways (Fig S5B). One of the barrier-strengthening p38 MAPKs, MAPK14/p38α, was also inferred by the TREKING model that describes barrier-disruptive signaling, highlighting the complexity of kinase functionality and kinase-mediated signaling during different phases of barrier perturbation. Consistent with model predictions, increased levels of activated kinases in ERK, JNK, and p38 pathways and the noncanonical, inferred kinase ABL1/c-Abl were detected within the first 15 min of thrombin treatment by Western blot (Fig S5C).

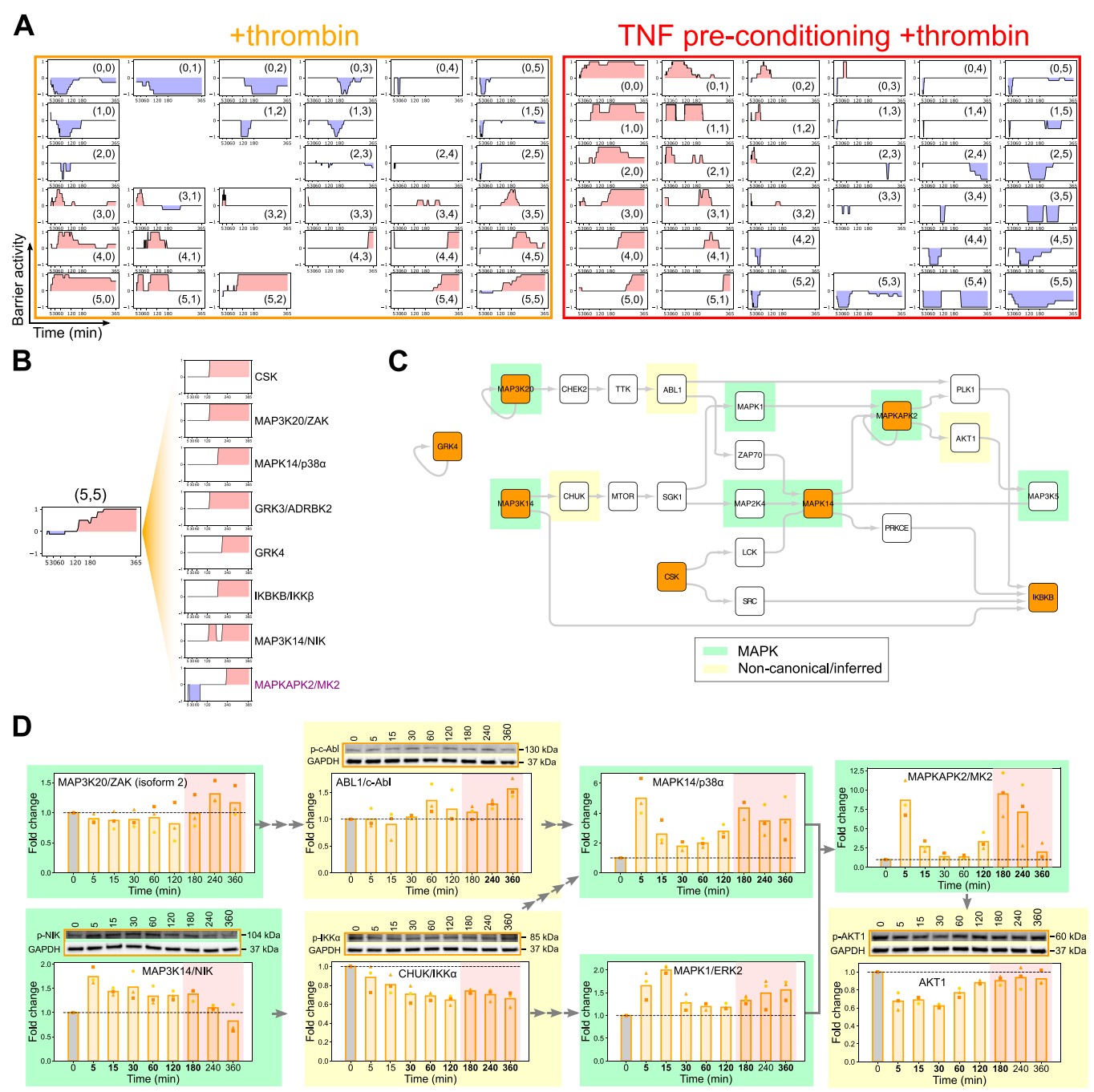

**Figure 6. TREKING maps alternate routes for activation of the MAPKAPK2/MK2 switch kinase.**
**(A)** SOM clustering of tKiR-predicted kinases with similar temporal functionality in +thrombin (left) or TNF preconditioning+thrombin (right) condition. Blue and red shadings represent barrier-weakening and barrier-strengthening functionality, respectively. In each neuron, the black line is the mean functionality of all the kinases within that neuron, with "+1" indicating all kinases are barrier-strengthening and "−1" indicating all kinases are barrier-weakening. Less than "1" means not all kinases in the neuron were predicted to be active in that time window. **(B)** Representative late barrier-strengthening-dominant neuron (neuron [5,5] in +thrombin condition) containing barrier-strengthening kinases of similar temporal barrier kinetics and switch kinase MAPKAPK2/MK2. **(C)** Local phosphosignaling network reconstructed from neuron (5,5) in +thrombin condition. Kinases predicted by tKiR are labeled in orange; TREKING-inferred upstream/downstream kinases are unfilled. MAPKs and noncanonical/inferred kinases are highlighted in green and yellow, respectively. **(D)** Phosphorylation of tKiR-predicted and TREKING-inferred kinases within the network in (C) was probed by Western blot in three biological replicates. Data were normalized to non-treated, media-only condition ("0," gray bars). Symbols represent the fold change of individual biological replicates, and bars represent the mean fold change of three biological replicates. Red shadings represent the late-stage (180–360 min after thrombin treatment) barrier-strengthening activity of neuron (5,5) as predicted by TREKING. Arrows represent the kinase connections in the network. The bolded timepoints indicate that the phosphorylation is different from the basal level (*P*-value from the *t* test below 0.05 or all biological replicates reporting a fold change increasing by at least 20% compared with non-treated cells). See also Fig S4, Tables S2 and S3.
Source data are available for this figure.

### MAPK14/p38α regulates the late activation of MAPKAPK2/MK2

Our models predicted that early and late barrier activity of MAPKAPK2/MK2 was regulated by distinct upstream MAPK kinases and that the late barrier strengthening activity in the thrombin-alone condition involved a MAP3K20/ZAK-MAPK14/p38α signaling pathway. To evaluate our predictions and ask if MAPK14/p38α is responsible for the increased activity of MAPKAPK2/MK2 in the barrier recovery phase, we used both genetic and pharmacological approaches (Fig 7A and Tables S4 and S5). We performed lentiviral knockdowns in HBMECs by targeting *MAP3K20* or *MAPK14*, and the knockdown efficiency was evaluated by quantitative reverse-transcription PCR (qRT–PCR) (Fig 7B; Table S4). We then measured the activity of MAPKAPK2/MK2 across 6-h time course after thrombin treatment by Western blot. Because knocking down MAP3K20/ZAK resulted in low cell viability and permeable barrier at the basal level, we did not pursue it further. Knocking down MAPK14/p38α eliminated the second wave of MAPKAPK2/MK2 activation compared with the scrambled control but preserved the first wave of MAPKAPK2/MK2 activation in the barrier disruption phase (Fig 7B), consistent with the model prediction that MAPK14/p38α contributes to the barrier activity of MAPKAPK2/MK2 primarily in the barrier recovery phase. We then used a small-molecule inhibitor, zunsemetinib, to selectively block the MAPK14/p38α activation of MAPKAPK2/MK2 (Wang et al, 2018) at 2 h after thrombin treatment. In the zunsemetinib-treated cells, there was an immediate decrease in the MAPKAPK2/MK2 phosphorylation mark as compared with the DMSO control (Fig 7C), consistent with the model that signaling from MAPK14/p38α is important for activating MAPKAPK2/MK2 in the barrier recovery phase.

Overall, results from both genetic knockdown and small-molecule inhibitor treatment support the model prediction that in +thrombin condition, MAP3K20/ZAK-MAPK14/p38α signaling contributes to the second wave of MAPKAPK2/MK2 signaling (Fig 7D). Our analysis also indicates that in +thrombin condition, both JNK and ERK signaling likely contribute to the early MAPKAPK2/MK2 activation, and the late barrier-strengthening pathway is rewired with TNF preconditioning (Fig 7D). In TNF preconditioning+thrombin condition, ERK and JNK signaling still both regulate early MAPKAPK2/MK2 activation, but TNF preconditioning increased early JNK activation (Fig 4). However, the late MAPK14/p38α-MAPKAPK2/MK2 barrier recovery pathway is reduced, and different p38 MAPK isoforms (MAPK11/p38β and MAPK13/p38δ) are predicted to play a more important role in barrier recovery in cells that are preconditioned with TNF (Fig 7D).

### tKiR and TREKING expand the current understanding of kinases and associated phosphosignaling in barrier regulation

Most previous and ongoing research has focused on certain stages of barrier perturbation or sampled sparsely during a time course. We therefore asked how TREKING predictions compared with the existing literature in their breadth and temporal resolution. To do this, we visualized portions of the phosphosignaling networks that had been reported in the literature to regulate the endothelial barrier in response to thrombin (Table S6). We then performed the same visualization with kinases that had been predicted by tKiR and finally, by

TREKING. Compared with the literature, TREKING substantially increased the number of kinases associated with thrombin-induced barrier regulation and was able to assemble most predicted kinases into time-resolved phosphosignaling networks (Fig 8; Video 1 and Supplemental Data 1). This highlights the power of TREKING to broadly dissect functionally important phosphosignaling in barrier regulation, with high temporal resolution.

## Discussion

Kinase signaling pathways are highly implicated in endothelial barrier regulation (Mehta & Malik, 2006; Kuppers et al, 2014; Komarova et al, 2017), and consequently, kinase inhibitors are being explored for treatment of vascular injury (Aman et al, 2012; Rizzo et al, 2015; Botros et al, 2020). However, the experimental tools for reconstructing the complex signaling pathways in cells are underdeveloped, which has hindered therapeutic applications. Here, we introduce TREKING, which uses a small 28-panel kinase inhibitor screen, coupled with dynamic xCELLigence-based barrier measurements to broadly interrogate kinase signaling pathways and to study the dynamics of barrier regulation.

In agreement with the previous work (Mehta & Malik, 2006), our results using tKiR implicated multiple MAPK signaling pathways in regulating thrombin-induced barrier changes. Because of the dynamic nature of the xCELLigence screen, it was possible to predict time windows of kinase barrier activity and in some instances to discern specific MAPK signaling cascades propagating specific barrier phenotypes. Furthermore, TREKING accurately predicted rewiring of phosphosignaling networks between the two inflammatory conditions. For instance, changes in both the early and late barrier activities of the ERK and the p38 pathways were validated by temporal Western blots. In addition, we used a combination of genetic knockdown and small-molecule inhibitor treatment to show that early and late MAPKAPK2/MK2 barrier activity was differentially regulated and that MAPK14/p38α controlled the late activation but not the early activity of MAPKAPK2/MK2 in the thrombin-alone condition. Beyond MAPKs, our findings highlight the complexity of endothelial barrier regulation with over 150 kinases being predicted to regulate thrombin-induced changes in barrier integrity.

The dichotomy between the signaling networks that activate MAPKAPK2/MK2 at early and late time points highlights the power of TREKING to identify and dissect complex signaling pathways regulating barrier phenotypes after different inflammatory stimuli. The stress-activated MAPKAPK2/MK2 is involved in cytoskeleton remodeling associated with barrier permeability stimuli (Ronkina et al, 2008). Activation of MAPKAPK2/MK2 leads to phosphorylation of LIM domain kinase 1 (LIMK1) and heat shock protein family B member 1 (HSPB1/HSP27), both of which regulate actin filament dynamics in endothelial cells (Gorovoy et al, 2005; Rada et al, 2021). Our analysis suggests that early barrier-weakening activity of MAPKAPK2/MK2 was controlled by the ERK and the JNK pathways and that late MAPKAPK2/MK2 activation was regulated by a MAP3K20/ZAK-MAPK14/p38α signaling pathway and crosstalk between p38, ERK, and JNK pathways. Notably, the small and large MAP3K20/ZAK isoforms are both activated by stress stimuli and have been implicated in p38-mediated regulation of actin

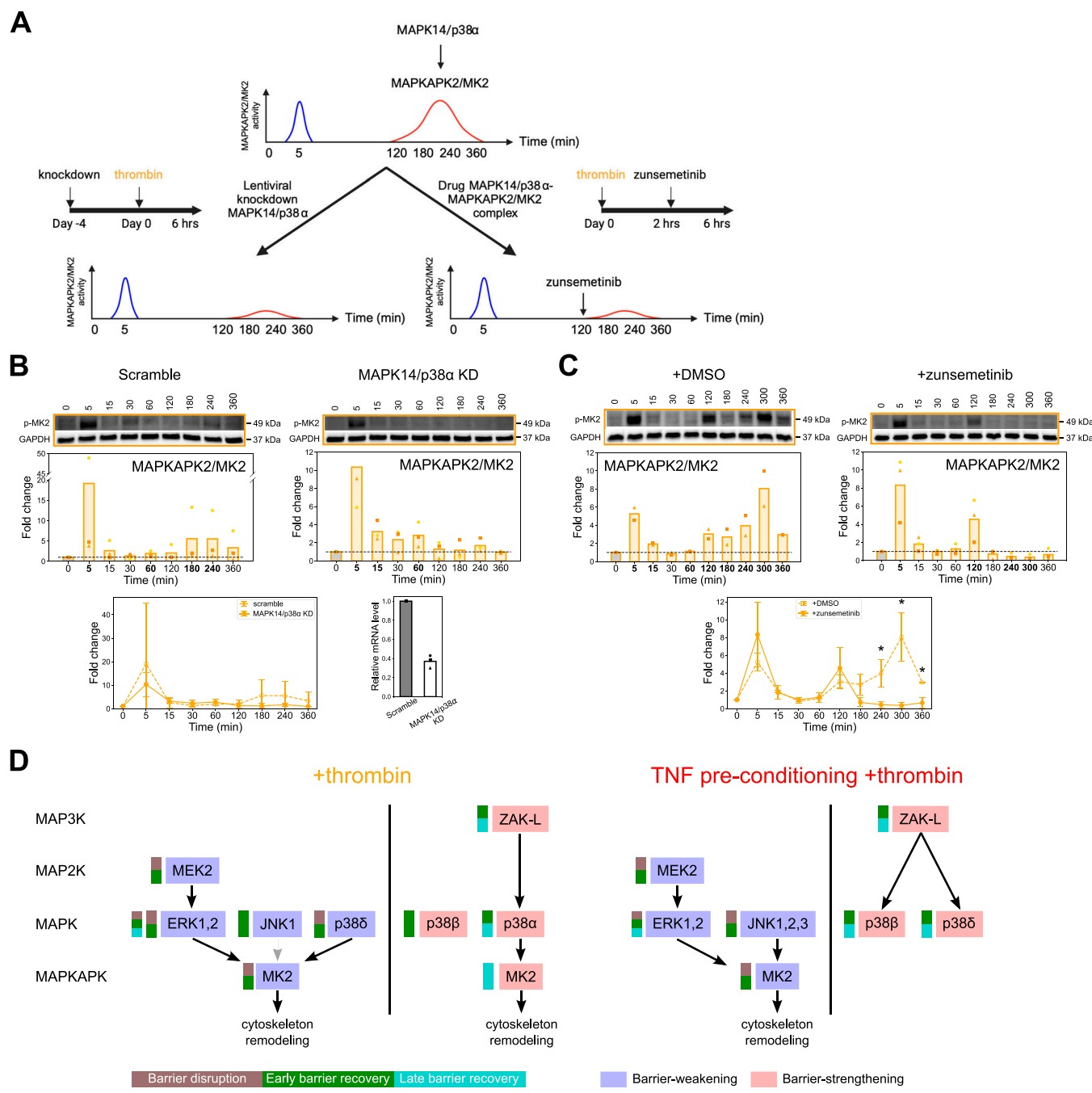

**Figure 7. MAPK14/p38α contributes to the late activation of MAPKAPK2/MK2 in the thrombin-alone condition.**
**(A)** Schematic showing the two approaches, that is, lentiviral knockdown and small-molecule inhibitor, used to validate the role of MAPK14/p38α in activating MAPKAPK2/MK2 in the barrier recovery phase. **(B)** Phosphorylation of MAPKAPK2/MK2 in scramble-transduced (top left) and MAPK14/p38α knockdown (top right) cells was probed by Western blot in three biological replicates. Data from these two graphs were plotted together as line graphs (bottom left, mean ± SD). MAPK14 expression was quantified by qRT–PCR (bottom right). Relative MAPK14 transcript abundance was determined by normalizing to the housekeeping gene GAPDH followed by normalizing to MAPK14 expression in the scrambled control sample according to the $2^{-\Delta\Delta CT}$ method. **(C)** Phosphorylation of MAPKAPK2/MK2 in DMSO- (top left) and zunsemetinib-treated (top right) cells was probed by Western blot in two (+DMSO) and three (+zunsemetinib) biological replicates. DMSO or zunsemetinib (10 $\mu$M) was added 2 h after thrombin treatment. Data from these two graphs were plotted together as line graphs (bottom, mean ± SD), and timepoint-wise comparison between DMSO and zunsemetinib treatment was analyzed by a $t$ test (*$P < 0.05$). Western blot data were normalized to non-treated, media-only condition ("0," gray bars). GAPDH was used as a loading control. Symbols represent the fold change of individual biological replicates, and bars represent the mean fold change of two or three biological replicates. The bolded timepoints indicate that the phosphorylation is different from the basal level ($P$-value from the $t$ test below 0.05 or all biological replicates reporting a fold change increasing or decreasing by at least 20% compared with non-treated cells). **(D)** Cartoon model summarizing kinase regulation of the MAPKAPK2/MK2 switch kinase in the two inflammatory conditions. "ZAK-L" stands for the larger isoform (isoform 2) of MAP3K20/ZAK. See also Tables S4 and S5.

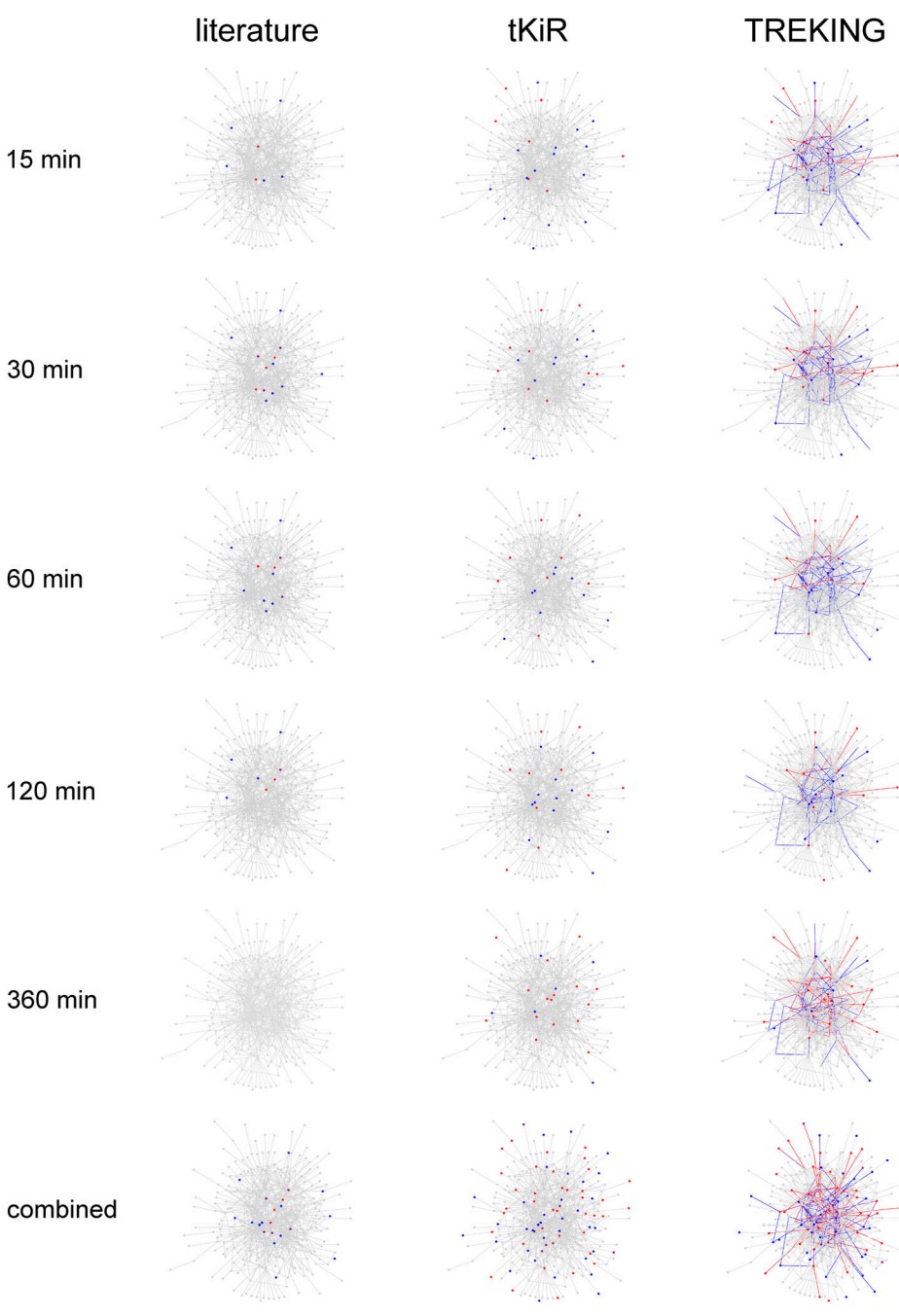

**Figure 8. tKiR and TREKING expand the current understanding of kinases and associated phosphosignaling in barrier regulation.** Comparison of literature-reported and model-predicted barrier-regulatory kinases and their connections at different stages of barrier perturbation. The background network in gray includes all kinase–kinase phosphorylation interactions from the kinase–substrate phosphorylation database PhosphoSitePlus. Blue and red nodes are the kinases reported by the literature or predicted by tKiR to have barrier-weakening and barrier-strengthening functionality, respectively; purple nodes are the kinases having conflicting literature reports. Blue and red edges are the interactions predicted by TREKING that are associated with only barrier-weakening and barrier-strengthening activity, respectively; purple edges are the interactions associated with both barrier-weakening and barrier-strengthening activities. The "combined" panel in the bottom is the composite literature reports or model predictions across all time points. The reported time points in the literature were expanded to 6-min time windows (3 min before/after the reported time points) to account for any experimental variations. See also Table S6 and Video 1.

stress fibers via different mechanisms (Gotoh et al, 2001; Nordgaard et al, 2022). Our analysis showed that the small and large MAP3K20/ZAK isoforms are phosphorylated with different kinetics after thrombin stimulation in HBMECs, suggesting another level of regulatory control. Together, our data suggest a complex regulation of MAPKAPK2/MK2 that is influenced by several MAPK pathways and is partially rewired by TNF preconditioning. Crosstalk between MAPK pathways in mediating barrier permeability has been implicated previously, but lack of molecular details describing their interconnections, associated temporal kinetics, and the role of noncanonical players makes it challenging to fully understand their regulatory mechanisms in the context of barrier

function (Paumelle et al, 2000; Surapisitchat et al, 2001; Zhang et al, 2001). These differences are often buried by conventional investigations that use tools such as genetic knockdown or knockout, which respond on the time scale of days rather than minutes. This highlights the importance of developing broad chemical screens, like TREKING, to systematically study the molecular details of kinase-driven signaling and to capture the differences in cell signaling between conditions.

Despite its power to elucidate kinase signaling with temporal resolution, TREKING is not without limitations. First, the kinase-compound biochemical data used for tKiR contains information only on a subset of the human kinome (291 of 518 kinases)

(Anastassiadis et al, 2011). Moreover, some kinases are targeted less frequently than others by the 28-kinase inhibitor panel, which can lead to both false-positive and false-negative predictions. To partially overcome this limitation, TREKING reconstructs phosphosignaling networks and infers kinases not predicted by tKiR and therefore can fill in missing gaps by leveraging kinase proteomic datasets. TREKING is also a discovery tool to identify new pathways that will need to be validated in the future. Second, PhosphoSitePlus, the knowledgebase used for building the phosphosignaling networks, despite being the most comprehensive catalog available to us, does not cover the complete kinase–substrate phosphorylation interactions in human cells nor is it specific to endothelial cells. As more advanced phosphoproteomics datasets and tools to dissect kinase–substrate relationships become available (Johnson et al, 2023), the subsequent iterations of TREKING will incorporate this comprehensive knowledge to further refine the models. Third, information, such as protein abundance, subcellular localization, translation, degradation, and dephosphorylation, was not incorporated in building TREKING models in the current study because of the lack of extensive knowledge on kinase abundance and kinetics in brain endothelial cells. The TREKING models can be further refined as this information becomes available.

In conclusion, TREKING allows (1) assessment of time-resolved kinase functionality associated with barrier regulation, (2) construction of phenotype-driving kinase-mediated phosphosignaling networks that recapitulate signaling events at different stages of barrier perturbation, and (3) systematic comparison of functional phosphosignaling networks between conditions. The global, time-resolved, and mechanistic details revealed by TREKING have ramifications beyond our fundamental understanding of phosphosignaling during inflammation. Beyond providing a deeper understanding into control of the vasculature, TREKING is broadly generalizable to other cellular systems with kinetic datasets and could be applied toward developing a molecular, kinetically resolved picture of other cellular phenotypes.

# Materials and Methods

## Cell lines

Primary HBMECs (Cat# ACBRI 376; Cell Systems) were cultured on rat tail collagen type I (5 $\mu$g/cm$^2$; Cat# 354236; Corning) in HBMEC culture media (Cat# CC-3202; Lonza) at 37°C and 5% CO$_2$. HBMECs were obtained at passage 3 and used until passage 9. HEK293-FT cells (Arang et al, 2017) were maintained in Gibco DMEM (Cat# 10569010; Thermo Fisher Scientific) supplemented with 10% FBS (Cat# F31016HI; SeraPrime), 25 mM HEPES, 2 mM L-glutamine (Cat# 25-005-CI; Corning), Gibco 1x MEM nonessential amino acids solution (Cat# 11140050; Thermo Fisher Scientific), and 100 $\mu$g/ml primocin (Cat# ant-pm; InvivoGen) at 37°C and 5% CO$_2$. HEK293-FT cells were used until passage 12.

## xCELLigence data acquisition

The kinetic data in this study are published and were acquired from xCELLigence assays using the xCELLigence Real Time Cell Analysis

Single Plate device (Agilent Technologies) (Dankwa et al, 2021). The assays are described here in brief. HBMECs were grown to confluency in xCELLigence 96 PET E-plates (Cat# 300600900; Agilent Technologies). On the day of the assay, HBMECs were equilibrated in serum-free culture media for 1–2 h, and thrombin (Cat# T6884; Sigma-Aldrich) was added at 5 nM. To capture thrombin-induced barrier disruption, the cell index was measured every minute for 6 min, after which kinase inhibitors were added in triplicate at 0.5 $\mu$M. The cell index was then measured every minute for 2 h and thereafter every 5 min for 4 h. The assays with TNF preconditioning were performed identically except that HBMECs were activated with 10 ng/ml TNF (Cat# 10291-TA; R&D Systems) for ~22 h before media equilibration. Data analysis was performed as described previously (Dankwa et al, 2021) with some modifications. The cell index was normalized to baseline (0) at the time point before addition of thrombin, and AUC was determined for each kinase inhibitor treatment. For this study, the AUC was determined within 5-min sliding windows over the 6-h time course. AUC values were normalized by subtracting the AUC of cells treated with thrombin+DMSO or TNF preconditioning+thrombin+DMSO. These values were then linearly transformed, with the most negative-normalized AUC value within a 5-min window being set to 0 and the normalized value for the control sample being set to 100.

## Lentivirus production

HEK293-FT cells were plated in 10-cm$^2$ tissue culture-treated dishes at $4 \times 10^6$ cells per dish and incubated at 37°C and 5% CO$_2$. The following day, cells were transfected at 70–80% confluency with 1.5 $\mu$g of pCMV-VSV-G envelope plasmid, 3 $\mu$g of psPax2 packaging plasmid, and 6 $\mu$g of pLKO.1 plasmid harboring shRNA against human *MAPK14* gene (Sigma-Aldrich MISSION shRNA library clones TRCN0000000509 and TRCN0000000511) or with a non-targeting scrambled shRNA (Sigma-Aldrich). 21 $\mu$l of 1 mg/ml polyethylenimine MAX (Cat# 24765; Polysciences) was mixed with pCMV-VSV-G, psPax2, and pLKO.1 in serum-free DMEM (Cat# 10569010; Thermo Fisher Scientific) and incubated for 10 min. Transfection mixtures were added dropwise to HEK293-FT cells. Media was changed the following day. Culture supernatant containing lentiviral particles was harvested ~24 h after the media change and stored at –80°C until use.

## Lentiviral transduction of HBMECs

Lentiviral transduction of HBMECs for knockdown of MAPK14/p38$\alpha$ was performed directly in the vessels at the time of cell seeding. 12-well plates (Cat# 3513; Corning) were coated with rat tail collagen type I (Cat# 354236; Corning) at 5 $\mu$g/cm$^2$ for 30 min at 37°C. HBMECs were seeded in 12-well plates at 40,000 cells/well with 1 mg/ml polybrene infection/transfection reagent (Cat# TR-1003-G; MilliporeSigma). Pooled lentiviruses were added to the corresponding wells at 0.5 ml per well. 24 h after transduction, puromycin (Cat# A1113803; Thermo Fisher Scientific) was added to the media at 2 $\mu$g/ml of final concentration. Cells were grown under puromycin selection for 2–3 d at which time cells were harvested for RNA extraction and TNF preconditioning for time course lysate collection was performed.

## Quantitative reverse-transcription PCR

RNA was extracted from TRIzol (Cat# 15596018; Thermo Fisher Scientific) homogenates by chloroform-based separation. Briefly, 80 µl of chloroform was added for every 400 µl of TRIzol reagent used. Samples were shaken gently for 15 s, incubated at room temperature for 3 min, and then centrifuged at 12,000$g$ and 4°C for 15 min. RNA contained in the upper aqueous phase was transferred to new tubes, 200 µl of isopropanol (Cat# T036181000; Thermo Fisher Scientific) was added for every 400 µl of TRIzol reagent used for RNA extraction, incubated at room temperature for 10 min, and then centrifuged at 12,000$g$ and 4°C for 10 min to pellet RNA. RNA pellets were washed three times using 75% ethanol (Cat# T038181000; Thermo Fisher Scientific), resuspended in 20 µl RNAase-free water, and incubated at 55–60°C for 15 min. cDNA was prepared from equal amounts of DNase-treated RNA of each sample using the iScript cDNA Synthesis Kit (Cat# 1708891; Bio-Rad) according to the manufacturer's protocol. qRT-PCR was performed using Power SYBR Green PCR Master Mix (Cat# 4367659; Thermo Fisher Scientific) and primers targeting human *MAPK14* gene (PrimerBank: https://pga.mgh.harvard.edu/primerbank/; forward primer: 5′-CCCGAGCGT-TACCAGAACC-3′ and reverse primer: 5′-TCGCATGAATGATGGACTGAAAT-3′). Samples were run on an Applied Biosystems QuantStudio 3 real-time PCR system (Cat# A28567; Thermo Fisher Scientific) with the following amplification conditions: a hold stage with 50°C for 2 min followed by 95°C for 10 min and a PCR stage with 40 cycles of 95°C for 15 s and 60°C for 1 min. Relative transcript abundance was determined by normalizing to the housekeeping gene GAPDH followed by normalizing to kinase expression in the scrambled control sample according to the $2^{-\Delta\Delta CT}$ method.

## Small-molecule inhibitor treatment

HBMECs were seeded in 12-well plates (Cat# 3513; Corning) at 35,000 cells/well and grown for 4 d. On the day of lysate collection, cells were equilibrated in serum-free culture media for 1 h and then treated with thrombin at a final concentration of 5 nM. Zunsemetinib (Cat# HY-139553; MedChemExpress) was added 2 h after thrombin treatment at a final concentration of 10 µM (0.1% DMSO). As controls, DMSO was added to another set of wells 2 h after thrombin treatment at a final concentration of 0.1%. Cell lysates were collected right before thrombin treatment and at 5, 15, 30, 60, 120, 180, 240, 300, and 360 min after thrombin treatment.

## Lysate preparation

HBMECs were seeded in six-well plates (Cat# 353046; Corning) at 55,000 cells/well and grown for 3 d. Cells were then activated with 10 ng/ml TNF for 21 h or kept in media for the same period. On the day of lysate collection, cells were equilibrated in serum-free culture media for 1 h and then treated with thrombin in triplicate at a final concentration of 5 nM for 5, 15, 30, 60, 120, 180, 240, and 360 min. After the indicated incubation periods, cells were washed twice with ice-cold PBS and lysed in SDS lysis buffer (50 mM Tris–HCl, 2% SDS, 5% glycerol, 5 mM ethylenediaminetetraacetic acid, 1 mM sodium fluoride, 10 mM $\beta$-glycerophosphate, 1 mM phenylmethylsulfonyl fluoride, 1 mM sodium orthovanadate, 1 mM dithiothreitol, supplemented with a cocktail of protease inhibitors [Cat# 4693159001; Roche] and phosphatase inhibitors [Cat# P5726; Sigma-Aldrich]). Cell lysates were clarified in filter plates (Cat# 8075; Pall) at 2,671$g$ for 30 min, after which they were stored at –80°C until use. Cell lysates from three biological replicates were collected.

## Western blot

All the gel electrophoresis was performed using Bolt 4–12% Bis-Tris mini protein gels. Proteins were transferred to PVDF membranes using the iBlot 2 (Cat# IB21001; Thermo Fisher Scientific) or iBlot 3 (Cat# A56727; Thermo Fisher Scientific) dry blotting system. Primary antibodies were used at concentrations recommended by the manufacturer (see Tables S3 and S5 for antibody information). The antibody to GAPDH (Cat# 97166, RRID AB_2756824; Cell Signaling Technology) was used as the loading control at 1:2,000 dilution. Blots were imaged using the Bio-Rad ChemiDoc imaging system, and signals were quantified using ImageJ2 (https://imagej.nih.gov/ij/, version 2.3.0). Background correction was performed for each band by subtracting background signals nearby the band. The signals from proteins of interest were first normalized to the signals from GAPDH, and then the signals at each time point were normalized to the signals at time zero for the fold change of phosphorylation from the basal level. Western blot was performed on cell lysates collected from three biological replicates.

## TREKING

To build TREKING models, we incorporated multiple methodologies, the details of which can be found in the methods that follow. Specifically, TREKING models are built by first employing elastic net regularization in a temporally resolved way that generates a list of predicted functional kinases within each time window. This generates a temporal trace for each predicted kinase, and these temporal traces are then organized into "neurons" using SOM methodology. To generate local networks for the kinases within each "neuron," we performed network generation steps, which resulted in the comprehensive TREKING models that are described within this study.

### Elastic net regularization

The elastic net regularization algorithm used for this study was published previously (Dankwa et al, 2021) but was adapted to generate time windows of kinase activity in barrier function. The KiR approach exploits the polypharmacology of kinase inhibitors and relies on linear combination of the contributions of kinases to cellular behavior (e.g., barrier permeability in this study) to make predictions on kinases important for specific cell phenotypes and the effect of untested kinase inhibitors on the phenotypes (Gujral et al, 2014). For the tKiR approach, a 5-min sliding time window was applied to the normalized cell index data, which slides at 1-min steps for the first 2 h after kinase inhibitor treatment and at 5-min steps afterward. For each time window, the algorithm was applied on the normalized AUC values and kinases with nonzero coefficients were predicted to be informative for barrier function. The

sign of the coefficients indicates the functionality of the kinases in regulating barrier integrity, with kinases having positive and negative coefficients being informed to have barrier-strengthening and barrier-weakening function, respectively.

### SOM

To group kinases that were predicted to have similar temporal barrier activity, the NumPy-based SOM implementation MiniSom (https://github.com/JustGlowing/minisom) was used to cluster the tKiR-predicted kinases by their temporal characteristics. A grid size of 6 × 6 was used to generate SOMs. For each SOM neuron, the barrier activity of kinases was plotted over time as a proportion of the kinases in that neuron. For example, if a neuron consisted of 4 kinases and all had the same predicted barrier activity during a specific time window, then they were assigned "+1" (if barrier-strengthening) or "−1" (if barrier-weakening), indicating that 100% of kinases had that same barrier activity. However, if only half of the kinases in that neuron (2/4 kinases) were active during a different time window, then it was assigned +0.5/−0.5 and so on.

### Network generation

To build the local phosphosignaling network for an SOM neuron, the shortest paths between any pair of kinases within that neuron was identified using the kinase–substrate phosphorylation database on PhosphoSitePlus as the background network, and all the paths were combined to form the local phosphosignaling network that describes the paths through which the signals may propagate with certain kinetics. Search for the shortest paths between kinases was performed using NetworkX (https://github.com/networkx/networkx), a Python package for analyzing complex network structures.

## Literature search

To compare the scope of previous research with the current study, a comprehensive literature search on protein kinases reported to regulate barrier function was performed using the free search engine PubMed (https://pubmed.ncbi.nlm.nih.gov). The search terms were "endothelial barrier thrombin" plus kinase gene names, and the search results were filtered so that only articles published on or after year 2000 were shown. The search was performed on each of the 518 human protein kinases. The search was not restricted to studies on HBMECs; instead, studies on thrombin-induced barrier disruption using endothelial cell lines, primary endothelial cells isolated from different organs, or in vivo studies using mice or rats were all included. To compare with our work, studies or data beyond a 6-h time window (after thrombin treatment) were excluded.

## Visualization

Schematics were created with BioRender.com. Phosphosignaling networks were visualized using Cytoscape (https://cytoscape.org, version 3.7.2), with the hierarchic layout from yFiles layout algorithms (https://www.yworks.com/products/yfiles-layout-algorithms-for-cytoscape).

## Statistical analysis

A $t$ test was used to evaluate the difference of kinase activity between non-treated and thrombin-treated conditions. SciPy statistical package (https://github.com/scipy/scipy, version 1.8.0) was used to perform the $t$ tests. At each time point, both $P$-value and fold change of phosphorylation with respect to non-treated cells were reported to determine if the two sets of data are different from each other. A $P$-value below 0.05 or all biological replicates reporting a fold change increasing or decreasing by at least 20% compared with non-treated cells indicates that the kinase activity is different between non-treated and thrombin-treated conditions.

# Data Availability

Data and code generated in this study are available in the main text or the supplementary information.

# Supplementary Information

# Acknowledgements

The authors would like to thank Mary-Margaret Dols for experimental guidance. This work was funded by National Institutes of Health (NIH) grants R01 AI148802, R61 HL154250, and R33 HL154250 (to A Kaushansky and JD Smith).

## Author Contributions

L Wei: conceptualization, data curation, software, formal analysis, validation, investigation, visualization, methodology, and writing—original draft, review, and editing.
S Dankwa: conceptualization, validation, investigation, methodology, and writing—review and editing.
K Vijayan: methodology.
JD Smith: conceptualization, supervision, funding acquisition, and writing—review and editing.
A Kaushansky: conceptualization, supervision, funding acquisition, and writing—review and editing.

## Conflict of Interest Statement

The authors declare that they have no conflict of interest.

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
