## [Reviewer comments · Life Science Alliance]

Life Science Alliance

Interrogating endothelial barrier regulation by temporally resolved kinase network generation

Alexis Kaushansky, Ling Wei, Selasi Dankwa, Kamalakannan Vijayan, and Joseph D Smith

DOI: <https://doi.org/10.26508/lsa.202302522>

Corresponding author(s): Alexis Kaushansky, Seattle Children's Research Institute

Review Timeline:

Submission Date:	2023-12-12
Editorial Decision:	2024-02-05
Revision Received:	2024-02-12
Editorial Decision:	2024-02-14
Revision Received:	2024-02-21
Accepted:	2024-02-22

Transaction Report:

February 5, 2024

Re: Life Science Alliance manuscript #LSA-2023-02522-T

Dr Alexis Kaushansky
Seattle Children's Research Institute
Center for Global Infectious Disease Research
307 Westlake Ave N
Seattle, WA 98109

Dear Dr. Kaushansky,

Thank you for submitting your manuscript entitled "Interrogating endothelial barrier regulation by temporally resolved kinase network generation" to Life Science Alliance. The manuscript was assessed by expert reviewers, whose comments are appended to this letter. We invite you to submit a revised manuscript addressing the Reviewer comments.

Thank you for this interesting contribution to Life Science Alliance. We are looking forward to receiving your revised manuscript.

Sincerely,

B. MANUSCRIPT ORGANIZATION AND FORMATTING:

Reviewer #1 (Comments to the Authors (Required)):

The manuscript by Wei et al describes a novel computational approach for prediction of specific roles for individual protein kinases and kinase networks in regulation of biological processes at a high temporal resolution. Authors use regulation of endothelial permeability as a model to interrogate the role of a broad panel of protein kinases in mediating this process downstream of thrombin signaling alone or in combination with TNF pre-stimulation. Authors convincingly demonstrate the advantages of the new approach and provide in-depth analysis of multiple parallel MAPK pathways that exhibit different functions at different timepoints following stimulation. Of particular interest is the detailed analysis of the dual function of MAPKAPK2/MK2 and the alternative upstream pathways regulating its activity. The manuscript is well written and experimental results support conclusions discussed by the authors. This work will be of interest to the readership of the journal as it describes a valuable technique applicable to a wide variety of biological studies. This reviewer has only minor comments that may improve the manuscript.

1. Line 139. The authors stated: "The correlation was the lowest in the early barrier disruption phase (Pearson's $r = 0.72$) and increased during early barrier recovery (Pearson's $r = 0.90$ to 0.92) (Fig 1D), suggesting that the phosphosignaling networks are more divergent during barrier disruption and more similar during the initial barrier restoration phase." It is possible that the correlation at the early phase is the lowest because inhibitors have different affinities for different kinases and thus have different inhibition kinetics. Authors should take this into account.
2. Figure numbers are not indicated on the pages with main figures.
3. Fig S5 has very poor resolution. It is hard to read some of the text labels, especially in S5B.

Reviewer #2 (Comments to the Authors (Required)):

The manuscript submitted by Wei et al. describes a method called Temporally Resolved Kinase Network Generation (TREKING), to investigate the temporal function and phosphosignaling networks of 17 key endothelial barrier-regulating kinases. The method is largely based on previous analyses already published (Dankwa, Cell Chem Biol, 2021) in which the authors use a panel of inhibitors to identify kinases involved in endothelial barrier disruption caused by proinflammatory signaling such as thrombin and TNF. The current manuscript is a follow-up and time-resolved refinement of the previous experimental approach. Using a 28-kinase inhibitor screen and machine learning, TREKING consist of using a 5-minute sliding time window, which slides at 1-minute steps for the first two hours after kinase inhibitor treatment and at 5- 160 minute steps afterwards. This allows the authors to identify the involvement of kinases at particular time-points after the stimulation of brain endothelial cells with thrombin and TNF. The authors focus on the MAPKAPK2/MK2 kinase exhibits dual activities-early barrier-weakening in inflammatory conditions and late barrier-strengthening exclusively with thrombin.

While lacking the technological novelty of the previous report, this manuscript advances and successfully stretches the capability of kinase regression to identify the temporal contribution of different kinases. The amount of work is substantial, the experiments are carefully executed, and there is sufficient validation of the results obtained using the panel of inhibitors.

Major point:

1. Please, report the differential effect of some of the pathways, preferable those involving MAPKAPK2/MK2 on a non-kinase effector (e.g. cytoskeleton regulators) that could justify the mechanism of action.

Minor points:

1. The schematic representation in Fig. 1A is confusing. The top line marking time periods of stimulation seems to mark the graph beneath because there is a blue area that connects one each other. It is not clear what such blue area means. It is necessary to go to Supplemental Figs. in order to understand these experimental settings. Please, reorganize and clarify this panel.

2. Fig 3 shows pathways as separated and unrelated, although, as the authors mention in the text and the common downstream kinases, there is a substantial overlapping between these pathways, clearly illustrated by MK2, which is controlled by more than one pathway. Please, reorganize in order to illustrate such overlapping, which is central to justify the chronological differences between the effector kinases and their upstream regulators.

We thank the reviewers for their constructive feedback that we feel has greatly improved the manuscript. Enclosed is a revised manuscript that addresses reviewers' comments. Below is a point-by-point response to the reviewers' comments.

Reviewer #1: The manuscript by Wei et al describes a novel computational approach for prediction of specific roles for individual protein kinases and kinase networks in regulation of biological processes at a high temporal resolution. Authors use regulation of endothelial permeability as a model to interrogate the role of a broad panel of protein kinases in mediating this process downstream of thrombin signaling alone or in combination with TNF pre-stimulation. Authors convincingly demonstrate the advantages of the new approach and provide in-depth analysis of multiple parallel MAPK pathways that exhibit different functions at different timepoints following stimulation. Of particular interest is the detailed analysis of the dual function of MAPKAPK2/MK2 and the alternative upstream pathways regulating its activity. The manuscript is well written and experimental results support conclusions discussed by the authors. This work will be of interest to the readership of the journal as it describes a valuable technique applicable to a wide variety of biological studies. This reviewer has only minor comments that may improve the manuscript.

We thank the reviewer for the positive appraisal of the TREKING methodology.

1. Line 139. The authors stated: "The correlation was the lowest in the early barrier disruption phase (Pearson's $r = 0.72$) and increased during early barrier recovery (Pearson's $r = 0.90$ to 0.92) (Fig 1D), suggesting that the phosphosignaling networks are more divergent during barrier disruption and more similar during the initial barrier restoration phase." It is possible that the correlation at the early phase is the lowest because inhibitors have different affinities for different kinases and thus have different inhibition kinetics. Authors should take this into account.

We agree with the reviewer that multiple factors, including differences in the kinetics of action of the kinase inhibitors, may reduce the correlation at the earliest time points. We added the following text to address this point:

"Of note, differences in the kinetics of action of the kinase inhibitors may also contribute to the reduction of the correlation at the earliest time points." (lines 140-142)

2. Figure numbers are not indicated on the pages with main figures.

We have added figure numbers on the pages with main figures.

3. Fig S5 has very poor resolution. It is hard to read some of the text labels, especially in S5B.

We thank the reviewer for bringing this up. We have replaced it with a higher resolution figure.

Reviewer #2: The manuscript submitted by Wei et al. describes a method called Temporally Resolved Kinase Network Generation (TREKING), to investigate the temporal function and phosphosignaling networks of 17 key endothelial barrier-regulating kinases. The method is largely based on previous analyses already published (Dankwa, Cell Chem Biol, 2021) in which the authors use a panel of inhibitors to identify kinases involved in endothelial barrier disruption caused by proinflammatory signaling such as thrombin and TNF. The current manuscript is a follow-up and time-resolved refinement of the previous experimental approach. Using a 28-kinase inhibitor screen and machine learning, TREKING consist of using a 5-minute sliding time window, which slides at 1-minute steps for the first two hours after kinase inhibitor treatment and at 5-160 minute steps afterwards. This allows the authors to identify the involvement of kinases at particular time-points after the stimulation of brain endothelial cells with thrombin and TNF. The authors focus on the MAPKAPK2/MK2 kinase exhibits dual activities-early barrier-weakening in inflammatory conditions and late barrier-strengthening exclusively with thrombin.

While lacking the technological novelty of the previous report, this manuscript advances and successfully stretches the capability of kinase regression to identify the temporal contribution of different kinases. The amount of work is substantial, the experiments are carefully executed, and there is sufficient validation of the results obtained using the panel of inhibitors.

We thank the reviewer for their positive review.

Major point:

1. Please, report the differential effect of some of the pathways, preferable those involving MAPKAPK2/MK2 on a non-kinase effector (e.g. cytoskeleton regulators) that could justify the mechanism of action.

We thank the reviewer for the thoughtful suggestion. We agree that it is important to look at the non-kinase effectors downstream of the kinases and kinase signaling pathways heavily studied in this work. The current manuscript focuses on identifying functional kinases and associated kinase signaling pathways on barrier regulation, and does not include non-kinase effectors regulated by functional kinases as part of this first version of TREKING. While beyond the scope of this manuscript, it is an interesting area of future study.

Minor points:

1. The schematic representation in Fig. 1A is confusing. The top line marking time periods of stimulation seems to mark the graph beneath because there is a blue area that connects one each other. It is not clear what such blue area means. It is necessary to go to Supplemental Figs. in order to understand these experimental settings. Please, reorganize and clarify this panel.

We thank the reviewer for pointing this out. The blue area in Fig 1A indicates the time period where thrombin treatment is on. This is the 6-hour time window we focus on in this study and is zoomed out from the top line which shows the entire stimulation including TNF pre-treatment as well. We have added another figure legend box inside the xCELLigence plot to clarify the meanings of the blue and gray shadings in Fig 1A.

2. Fig 3 shows pathways as separated and unrelated, although, as the authors mention in the text and the common downstream kinases, there is a substantial overlapping between these pathways, clearly illustrated by MK2, which is controlled by more than one pathway. Please, reorganize in order to illustrate such overlapping, which is central to justify the chronological differences between the effector kinases and their upstream regulators.

We thank the reviewer for their helpful comment. To address this point, we added black dots by the kinases that overlap between these MAPK pathways, i.e., CSF1R, TAOK1, and MAPKAPK2/MK2, and noted in the figure legend.

February 14, 2024

RE: Life Science Alliance Manuscript #LSA-2023-02522-TR

Dr. Alexis Kaushansky
Seattle Children's Research Institute
Center for Global Infectious Disease Research
307 Westlake Ave N
Seattle, WA 98109

Dear Dr. Kaushansky,

Thank you for submitting your revised manuscript entitled "Interrogating endothelial barrier regulation by temporally resolved kinase network generation". We would be happy to publish your paper in Life Science Alliance pending final revisions necessary to meet our formatting guidelines.

- please be sure that the authorship listing and order is correct
- LSA allows supplementary figures, but not EV and Appendix Figures; please update your callouts for the Supplementary Figures in the manuscript Appendix Fig S1 = Fig S1A). Data sets should be renamed to supplementary tables, and the callouts should read Table S1, etc.
- please remove the file called appendix text. Supplementary figures should be uploaded separately, and their legends should be provided in the manuscript text after the references
- please add your main, supplementary figure, table, and video legends to the main manuscript text after the references section
- please add the Twitter handle of your host institute/organization as well as your own or/and one of the authors in our system
- there is a discrepancy in the presentation of the name of your co-author -- please correct Joseph D Smith in the manuscript file vs. Joe Smith in the system
- we encourage you to revise the figure legends for Figure 4 such that the figure panels are introduced in alphabetical order
- please add callouts for Figures S1A-B and S3B to your main manuscript text

A. FINAL FILES:

B. MANUSCRIPT ORGANIZATION AND FORMATTING:

Sincerely,

February 22, 2024

RE: Life Science Alliance Manuscript #LSA-2023-02522-TRR

Dr. Alexis Kaushansky
Seattle Children's Research Institute
Center for Global Infectious Disease Research
307 Westlake Ave N
Seattle, WA 98109

Dear Dr. Kaushansky,

Thank you for submitting your Methods entitled "Interrogating endothelial barrier regulation by temporally resolved kinase network generation". It is a pleasure to let you know that your manuscript is now accepted for publication in Life Science Alliance. Congratulations on this interesting work.

DISTRIBUTION OF MATERIALS:

Again, congratulations on a very nice paper. I hope you found the review process to be constructive and are pleased with how the manuscript was handled editorially. We look forward to future exciting submissions from your lab.

Sincerely,
